# Development and Piloting of a Web-Based Tool to Teach Relative and Absolute Risk Reductions

**DOI:** 10.3390/ijerph192316086

**Published:** 2022-12-01

**Authors:** Sandro Zacher, Birte Berger-Höger, Julia Lühnen, Anke Steckelberg

**Affiliations:** 1Institute of Health and Nursing Science, Martin Luther University Halle-Wittenberg, 06112 Halle (Saale), Germany; 2Institute for Public Health and Nursing Research, University of Bremen, 28359 Bremen, Germany

**Keywords:** health literacy, evidence-based medicine, education, risk assessment, numerical data

## Abstract

Interpreting study results is an essential component of decision-making. Both laypeople and healthcare professionals often misinterpret treatment effects that are presented as relative risk reduction. Therefore, we developed and piloted a web-based tool to teach the difference between relative and absolute risk reductions. This project follows the UKMRC-guidance for complex interventions. The tool was developed based on adult learning and design theories. This was followed by a qualitative feasibility study focusing on acceptance, applicability, and comprehensibility with healthcare professionals and laypersons. We conducted think-aloud and semi-structured interviews and analysed them using qualitative content analysis. In addition, we explored calculation skills. Between January 2020 and April 2021, we conducted 22 interviews with 8 laypeople and 14 healthcare professionals from different settings. Overall, the tool proved to be feasible and relevant. With regard to comprehension, we observed an awareness of the interpretation of risk reduction, presented therapy effects were questioned more critically, and the influence of relative effects was recognized. Nevertheless, there were comprehension problems in some of the participants, especially with calculations in connection with low mathematical skills. The tool can be used to improve the interpretation of risk reductions in various target groups and to supplement existing educational programs.

## 1. Introduction

The interpretation of study findings on treatment options is an essential component in decision making about health issues and often challenges people in decision making. Not only laypeople but also healthcare professionals find it difficult to interpret the effects of treatments. When compared with absolute risk reduction, treatment effects presented as relative risk reduction were perceived as larger and more convincing by both laypeople and healthcare professionals [1,2]. The relative risk reduction describes the difference between the event rates (risk) in the control group and the intervention group in relation to the event rate in the control group (baseline risk). If the risk of an event is 10% in the control group and 5% in the intervention group, the relative risk reduction is 50%. The relative risk reduction thus expresses the proportion by which an event can be reduced by the intervention. The problem: The relative risk reduction is also 50% if the event is 2% in the control group and 1% in the intervention group. Thus, no direct benefit for the individual person can be derived. Moreover, the usually larger number appears more impressive than the absolute risk reduction. The absolute risk reduction is the difference between the risk in the control group and the risk in the treatment group. If the risk of an event is 10% in the control group and 5% in the intervention group, the absolute risk reduction is 5%. In other words, in 5 out of 100 people, the event does not occur as a result of the intervention.

The influence of the risk presentation manifests itself, for example, in the more frequent prescription of medications, the benefits of which have been presented as relative risk reduction or in a greater tendency to treat hypertension in patients with a described relative risk reduction compared to an absolute risk reduction [3,4]. The necessity of a critical assessment of health information can be illustrated using the thrombosis risk of oral contraceptives. In 1995, there was a warning in the UK that third-generation oral contraceptive pills increase the risk of life-threatening blood clots by 100%, i.e., they double the risk. The relative risk reporting led to fear among women and the withdrawal of the pill with the result that abortions from unwanted pregnancies increased and a distrust of oral contraceptive pills arose. Looking at the absolute values, the risk of thrombosis was 1 in 7000 women for the second generation of contraceptive pills and increased to 2 in 7000 women for the third generation of contraceptive pills. It is questionable whether communicating the absolute increase in risk would have led to a similar reaction [5].

In this context, the numeracy skills of the individuals concerned play a major role [6]. The correct presentation of benefits and harms of a treatment and the critical health literacy [7] of the users is the prerequisite for participation and informed decision-making [8,9]. Considering the influence of the presentation of the relative risk reduction, these prerequisites are not given, and participation cannot be realized.

The “Informed Health Choices (IHC)” project provides support for a critical appraisal of treatment allegations. For this purpose, key concepts have been defined, which can be used to assess the credibility of health claims [10,11]. A study published in 2016 determined 415 interventions that conveyed at least one of the key concepts. The topic “Weigh benefits and harms” was most frequently addressed (273 articles). Noticeably fewer articles (9) dealt with the topic “Relative effects can be misleading” [12]. In 2017, the Network for Evidence-based Medicine defined the contents of educational offers for professionals in healthcare and for the public with the curriculum “Evidence-based decision-making”, which is intended to enhance critical health literacy [13]. In addition to these approaches, a more recent study has shown that patients and healthcare professionals alike prefer relative effect measures in decision-making [14]. Possible influences of advertisements for treatments and therapies as well as the current discussion on the communication of the vaccine efficacy of Covid vaccines highlight the relevance of understanding the difference between relative and absolute risk reduction [15]. Laypeople and health professionals are difficult to reach outside institutional learning environments, but are exposed to misleading representations of treatment effects in their everyday or professional health decisions.

In the present study, therefore, a web-based tool has been developed to convey the difference between relative and absolute risk reductions and to increase critical health literacy. This study aimed at exploring the feasibility of a web-based tool to provide self-directed learning of relative and absolute risk reductions in the interpretation of treatment effects for both laypeople and healthcare professionals.

The project was conducted according to the UK MRC-guidance for complex intervention, focusing on the phases development and piloting (Phases I and II) [16]. The reporting guideline CEeDECI 2 [17] was used for the documentation and the developed intervention was described according to the “Guideline for reporting evidence-based practice educational interventions and teaching” (GREET) [18]. The COREQ checklist was used for the documentation of the qualitative methods [19].

## 2. Phase I: Development of the Educational Intervention

### 2.1. Methods

#### 2.1.1. Systematic Literature Search

In March and April 2019, a systematic literature search was carried out in the databases Medline via PubMed, PsycInfo and Psyndex via PubPsych to identify and analyse existing educational tools. This was followed by an internet search in Google and databases for learning resources. The database search was supplemented by backward citation tracking and queries into known projects. Following Albarqouni et al., [20] data extraction was based on general survey characteristics as well as on the characteristics of the participants, the intervention and the evaluation. Further information about the literature search can be found in Appendix A.

#### 2.1.2. Theoretical Design

The theoretical framework used for the educational intervention was Knowles Adult Pedagogy [21]. According to this, the relevance of the topic should be made perceptible to the learners, for example by pointing out gaps in their knowledge. Furthermore, the active involvement of the learners should be achieved through self-controlled learning experiences and the transfer of responsibility for learning. According to Knowles, a direct relationship to situations in life is also helpful in order to present possible applications and to increase motivation through the prospect of personal development [21].

Presenting the contents in the form of e-learning, i.e., using the internet to impart knowledge and abilities [22], offers several advantages with regard to the requirements of the target group. For a start, accessibility and flexibility are higher, which makes integration in everyday working life easier and, in addition, it is possible to personalize the contents [23]. The learning time and pace and the consolidation of the contents can be individually adapted to the requirements [24]. Likewise, contents can be conveyed in a standardized way due to the web-based instruction form, allowing quick updating [22]. 

The instruction development was based on the ADDIE (**A**nalyze, **D**esign, **D**evelop, **I**mplement, and **E**valuate) approach by Branch [25], including the analysis of the target group and the content as well as the development of the didactic procedure and the definition of objectives. The target group consists of medical laypeople and healthcare professionals, who are in some way involved in the decision-making process, as well as multipliers such as healthcare teachers. Heterogeneous medical, statistical and scientific prior knowledge was to be expected in the target group, so the level of content was therefore adapted to suit medical laypeople. On the basis of the preliminary investigations into how the presentation form of risk reduction can have an influence and how informed decision making can be supported, learning objectives according to Bloom’s taxonomy [26] (Figure 1) and corresponding strategies for reviewing the learning target were developed along with didactic strategies.

In addition, the ARCS (**A**ttention, **R**elevance, **C**onfidence and **S**atisfaction) approach for motivational learning design by Keller [27] was integrated into the instruction development, which should help to make the tool’s learning objectives both appealing and interesting. With the help of the problem-oriented design process, the target group and existing training material were analysed using four categories of human motivation: attention, relevance, confidence and satisfaction and assumptions about the initial motivation of the trainees were formulated [27]. Due to low prior knowledge and lack of or little professional contact with the topic of relative and absolute risk reduction, the attention and perceived relevance of the majority of the target group were assessed as being rather low. Self-confidence with regard to statistical themes was also assessed as being low, but it was also expected that some of the trainees would overestimate their own abilities. Acceptable baseline satisfaction was anticipated. The assumptions about motivation were then used to define motivational goals and develop motivational strategies, as well as to integrate these strategies into instructional development using the ADDIE approach [25].

The “Cognitive Theory of Multimedia Learning” [28] was applied in designing the tool, combining the principles of cognitive stress [29], the doubled coding of information [30] and the active processing [31]. Furthermore, the selection of the forms of presentation within the tool was based on the recommendations of the guideline for evidence-based health information [32].

### 2.2. Results

#### 2.2.1. Systematic Literature Search

We were able to identify three evaluated, respectively, partially evaluated educational interventions [33,34,35]. However, none of these interventions could be adapted due to inadequate investigation of the influence exerted by the presentation form of risk reductions [33,34], to the focus on a specific disease [35] and to the lack of interaction possibilities for the trainees [33,35]. The tools identified in the internet search showed a high degree of heterogeneity with regard to the target group and the objective. Similarly, the critical examination of the influence of relative risk reductions was only contained in a few applications. An overview of the identified tools can be found in the Appendix A. All in all, none of the identified learning programs could be applied for the objectives of the project, which meant that the tool had to be fundamentally redeveloped. Nevertheless, a few approaches of the applications found through the search were able to be used as inspiration for the newly developed tool.

#### 2.2.2. First Concept of the Tool

Based on the preliminary considerations, the content to achieve the learning objectives was created and the first version of the tool was designed. The core aspect of the concept is that the trainees should be in charge of their own learning experience and the learning outcome. Starting with their own experiences, the participants should thus be enabled to establish a direct reference to professional and/or personal application possibilities, thereby increasing the perceived relevance of the topic. In addition to the sequence of topics provided by the tool (Figure 2), a direct selection of the contents is possible via a menu.

On the home page a small section can be found concerning the tool’s objectives and an explanation about the navigation and interaction possibilities. The thematic introduction uses two problem situations (PS) in which the trainees have to solve tasks. In the first everyday problem situation (PS1), two price-reduced yoghurts are used to explain the difference between an absolute and a relative price reduction, whereby the trainees have to select the yoghurt with the largest price reduction. The second problem situation (PS2) makes a connection to a health topic, using cancer screening as an example. Based on the two risk reductions shown, the trainees have to select the information that convinces them more of the importance of cancer screening. Presented are the absolute (0.1%) and the relative (25%) risk reduction in the sigmoidoscopy in the early detection of colon cancer [36]. The fact that the risk reductions presented relate to one and the same screening examination is only revealed to the participants as part of the answer, thus illustrating a possible influence.

The type of the screening examination was kept open for the trainees as long as possible in order to allow for a contact point with many participants. The reference to sigmoidoscopy only becomes apparent in the conclusion (C). Using cancer screening as an example, the tool demonstrates the calculation of absolute and relative risk reduction. For this purpose, the principle of a randomized controlled trial to test the effectiveness of the screening examination is explained with the help of animated pictograms (IS).

The calculations of the relative and absolute risk reduction are identically structured (RRR/ARR). Animated pictograms are used to describe the calculation verbally and no complex formulas are applied in order to take people with low mathematical understanding into consideration. As an option, 2 × 2 tables with a comprehensive derivation can be called up. The calculation can be repeated and consolidated with the use of optional exercises (TYK). The calculations are based on event rates represented in pictograms and can be performed, checked and corrected directly in dialog boxes within the tool. If necessary, the formula for the calculation can be called up.

In the conclusion (C), the trainees are given a final result referring to the influence of the relative risk reduction and two further in-depth options (PE/BR) as well as the sources (Q) and further links (RL). In one of the in-depth options (PE), the effectiveness of a new medication for reducing cardiovascular mortality compared with the standard medication [37] is calculated on the basis of the absolute risk reduction and is then compared with an advertisement stating a 20% lower cardiovascular mortality. For the sake of simplicity, the hazard ratio of 0.8 was presented as a relative reduction of 20%.

In the second in-depth option (BR), the significance of the baseline risk is clarified in line with the effect of statins for the primary prevention of heart attacks. The effect of statins in primary prevention is independent of different risk factors comparably constant and is relatively reduced by about 20% [38]. By using an interactive bar chart, the trainees can see the various absolute risk reductions depending on the baseline risk with a constant relative risk reduction and transfer them to a table.

The educational intervention design was according to the Cognitive Theory of Multimedia Learning and the recommendations derived from this [28]. The given information was double coded by means of text and graphic and related information was presented as closely together as possible. Illustrations and animations were preceded as far as possible by an explanatory text for better orientation. Pictograms and bar charts were used for the graphic presentation of the risk reductions as recommended in the guideline for evidence-based health information and the content reduced to the essentials [32]. However, in view of the heterogeneity of the target group and the different initial knowledge, it had to be assumed that not all the participants were familiar with basic terminology so that key terms were optionally explained using tooltips. To take the cognitive capacity of the trainees into account, the contents were divided into individual, separate sequences and the basic difficulty of the examples and explanations was set at the presumed level of the medical laypeople. The participants are also able to adjust the complexity and the expansion of the information content independently, for example by displaying 2 × 2 tables or inserting and removing the formulae for the calculation. Furthermore, the complexity of the possible in-depth options increases.

The programming was performed by a student (K.B.) in cooperation with the Department of Computer Science and Media at the Brandenburg University of Applied Sciences and allows the tool to be used with any modern internet browser, regardless of the end device. The text was edited with the support of a medical journalist (I.H.).

## 3. Phase II: Piloting of the Tool

### 3.1. Methods

Piloting was carried out using a qualitative feasibility study concerning acceptance, applicability and comprehensibility in an iterative process of analysis and revision.

#### 3.1.1. Setting and Sample

Suitable participants were men and women over 18 interested in health matters and who understood and spoke the German language. They should also have basic knowledge about using websites on their own. Participants who already had in-depth and application-oriented knowledge on the interpretation and calculation of risk reductions were ineligible for this study, but statistical knowledge gained during the study of medicine was not seen as an exclusion criterion. It was intended for the sample to include interested laypeople and professional people from the health sector; in particular, physicians were planned as direct participants in the decision-making processes and teachers in the healthcare system were planned as multipliers. In addition, other professionals with direct or indirect contact to patients were also eligible and a balanced relationship between age and gender and, in the case of laypeople, with regard to educational qualifications was aimed for. The first piloting phase was planned with four physicians, four teachers and four laypeople. This was followed by an interactive procedure with revisions until information saturation was reached. The recruiting of professionals in the health sector was carried out by approaching existing contacts to education and training institutions. At the time of the study, SZ was working as a teacher for health and nursing at one of the educational institutions; consequently, no recruiting took place in this special field. Laypeople were recruited via an advertisement in Ebay Classifieds. Due to problems in the recruiting of laypeople, an expense allowance for all participants was subsequently provided for all participants.

#### 3.1.2. Data Collection

The data collection was a multi-step process with qualitative and quantitative elements and was carried out by S.Z. and two student assistants (N.S., K.M.) (first semester in the Master’s program Health and Nursing Science). For the purpose of feasibility testing, the acceptance, applicability, comprehensibility and explorative efficacy regarding knowledge were investigated. Acceptance was defined as the reaction of the participants to the methodological implementation. Applicability concerned the practical application of the tools by the participants with regard to structure, handling and design. Comprehension was considered from the perspective of the participants as well as from the objective of the tool, both in terms of content and application. The evaluation of acceptance, applicability and understanding was carried out during the processing of the tool using the concurrent think-aloud method [39]. The aim was to observe the reactions of the test persons to the individual elements of the tool and to derive signs of feasibility or potentials for change. The think-aloud procedure was standardized (Appendix B) and at the same time field notes were made. A semi-structured interview took place directly afterwards. This served first to validate ambiguous statements made by the trainees and then to set the focus on aspects of the tools that might not have been mentioned until then. The key questions for this were developed in advance and discussed in the team (Appendix C). The explorative efficacy of knowledge was carried out with the Critical Health Competence Test (CHC Test) [40], which is a validated instrument for measuring critical health literacy consisting of four medical scenarios with a total of 72 items. To assess the knowledge, two items with increasing difficulty were recorded before and after the tool was used (Appendix D). For the description of the sample, the participants filled out a questionnaire with details to gender, age, educational level, profession and prior knowledge about the topic. They also gave a self-evaluation about their knowledge of mathematics and computer literacy, using a scale of 0–10. The survey was anonymous and to ensure the anonymity the participants set up their own four-digit code according to given criteria. The data collection took place during January and February 2020 in the rooms of the education and training institutions. Due to the SARS-CoV-2 pandemic, there was a pause in the data collection from March 2020 until November 2020 when it was continued using the Webex platform [41].

#### 3.1.3. Data Analysis

The evaluation of the socio-demographic data and the CHC items was performed in a descriptive manner. The analysis of the audio-recordings and transcriptions of the think-aloud protocols as well as the semi-structured interviews were carried out by means of a qualitative content analysis according to Mayring [42] carried out by S.Z., B.B.-H. and J.L. Because of the rising information saturation after the final revision, a transcription of the last 10 interviews was waived, analysing only the audio-recordings. The software MAXQDA [43] in the version of 2018 and Excel was used for the analysis. Based on the research question, a category system with main and subcategories was deductively created in accordance with the components of the feasibility testing described by Bowen and colleagues [44]. Furthermore, this category system could be flexibly adapted and extended during the analysis (see Appendix A). With the help of an iterative process consisting of piloting, analysis and revision, it was intended to achieve theoretical data saturation. The transcription and the findings were not presented to the participants for correction and confirmation due to the additional organizational work involved.

### 3.2. Results

#### 3.2.1. Sample

Between January 2020 and April 2021, 22 interviews with 8 laypeople and 14 professionals from various sectors of the healthcare system were carried out. The healthcare professionals were five physicians, four teachers, three health and nursing science students, one trainee geriatric nurse and one manager for case management. Their characteristics are shown in Table 1.

#### 3.2.2. Qualitative Feasibility Study

The data collection took around 70 min on average (range 36–94). Processing the tool using think-aloud took around 31 min on average (range 10–60). Theoretical data saturation was assumed since no further new findings could be gained. By means of the qualitative content analysis, four main categories with subcategories were created: 1. Acceptance, 2. Applicability, 3. Comprehension and 4. Achievement of didactic objectives. Quotes for characteristic examples can be found in Table 2 and were labelled with “L” for laypersons and “P” for professionals and were linguistically adapted.

Acceptance

This category describes the satisfaction, the emotional reaction, the perceived relevance and the potential benefit, respectively, recommendation as a reaction to the methodological implementation of the tools.

Satisfaction: The participants reported that they were generally satisfied with the tool, which was mainly due to a perceived increase in knowledge or a revision of knowledge (Table 2 Q1). Dissatisfaction arose before the tool was revised due to the lack of in-depth options and the presentation of the calculation using the 2 × 2 table. Furthermore, two professionals were dissatisfied due to unfulfilled expectations. For example, one person expected a tool for calculating risk reductions from studies.

Emotional reaction: The tool triggered both positive and negative emotions in the participants. Both laypeople and professionals found the mathematical content of some of the tool’s sectors to be overwhelming. This led in some cases to resignation and dispensing with assignments (Table 2 Q2). The participants named their own knowledge of mathematics as the reason for this. Nevertheless, particularly the simple introduction and repetition tasks had a motivating effect and led to an “aha effect” (Table 2 Q3).

Relevance: The content and objectives of the tool were assessed as relevant by the participants due to the lack of awareness and confrontation with the issue so far, as well as the professional and everyday contact (Table 2 Q4). Some of the participants reported that the tool is relevant for personal decision-making; others, however, did not find the tool to be relevant for their own decision-making (Table 2 Q5). The participants described realistic practical applications and a low level of difficulty as factors that promoted relevance. On the other hand, some sections were considered to be less relevant if a mathematical scientific approach had been taken and the participants already thought they had understood the content.

Benefit/Recommendation: Some participants saw the application of the tool on the one hand in the training of people in the health sector, and as an offer for the general public as well as personal further training. Laypeople in particular, on the other hand, felt that the application was only useful for those interested and for a specific purpose.

2.Applicability

This category contains statements about the practical use with regard to structure, handling and design of the tool and its contents.

Structure: It was possible to identify facilitating and hindering factors for applicability with regard to the structure. The participants found the recognizable structure of the tool and the combination of graphics and texts to be facilitating as well as the reduction in the content. Hindering factors were identified in the form of the unclear classification of texts and graphics and confusing menu displays. Due to the adaptation of the contents to the level of knowledge of laypeople, more in-depth contents, such as 2 × 2 tables or further technical terms, were presented as optionally accessible elements. This was not recognized in some cases. In addition, before the tool was revised the participants noted a greater need for opportunities to recapitulate (Table 2 Q6). Being able to choose which type of risk reduction one should begin with was not perceived by the participants as a relevant decision.

Handling: Regarding the handling of the tool, the participants found using well-known interaction and navigation features such as “help” and “next” buttons to be beneficial. They also described the possibility to do calculations and corrections directly in the tool and showing the solutions as being very helpful (Table 2 Q7). Although the integrated calculator was generally seen as being positive, the fixed calculation path caused problems for participants who used a different formula or none at all. Before the revision, a major problem in the operation was the section on baseline risk (BR). The use of the interactive bar chart as well as the corresponding exercise only became clear to the participants when they received a hint.

Design: The participants considered the tool design and the form of individual components mostly as helpful for understanding the contents. The graphics were described mostly as being clear and comprehensible. By using well-known design elements, the contents could be processed easily. Furthermore, the design contributed for the most part to the transfer of knowledge. The use of animations was rated in different ways. These were partly described as facilitating the spatial imagination and imparting the contents, but some important components were not noticed and the contents thus not clear.

The design of the interaction option before revision took place led to the fact that some of the participants did not perceive symbols as interaction options, which meant that, for example, no help could be called up (Table 2 Q8).

3.Comprehensibility

The category consisted of various aspects of comprehending and understanding as a prerequisite for comprehension from the participants’ perspective, compared with the tool’s objectives. It also contains the subcategories of understanding the content, difficulty of the exercises, text difficulty and transferability.

Understanding the content: Taking the entire tool into consideration, the participants described a heightened awareness of the different ways in which risk reductions are presented (Table 2 Q9). Some of the professional participants described the contents as a deepening or repetition with no additional knowledge gain. Participants recognized that the absolute risk reduction was rather low; however, attitudes toward screening were generally very positive (Table 2 Q10). Moreover, some of the participants interpreted the contents presented before the tool was revised more as information on cancer screening than on risk reduction. The participants generally described exercises and the use of natural frequencies as conducive to understanding. Different levels of understanding of the individual sections of the tool were observed. Some of the participants said the individual sections were understandable, others said they were not. The introductory content was predominantly regarded as comprehensible, while increasing difficulties were observed with the mathematical content and calculations. Furthermore, the section on baseline risk (BS) was mostly not understood before revision took place.

Difficulty of the exercises: The difficulty of the exercises and repetition tasks was described by the participants in very different ways; an increasing difficulty could be observed after the revision. The participants found the use of simple figures and the optional presentation of formulae to be useful, whereas missing graphics and unclear formulation and complexity of the exercises led to difficulties. Another factor for the difficulties were the calculations and the associated mathematical skills (Table 2 Q11).

Text difficulties: The participants reported having difficulties with the text due to the complexity and quantity (Table 2 Q12). Laypeople also had problems with terminology such as “cardiovascular”, “event rate” and “intervention group”.

Transferability: Both laypeople and professionals reported that they wanted to raise the awareness of the difference between relative and absolute risk reduction and their significance in everyday practice (Table 2 Q13). Some of the participants saw the retentiveness and the influence on the decision critically (Table 2 Q14).

4.Achievement of didactic objectives

This category contains descriptions that can be traced back to the didactics of the tool. In some cases, factors already described led to the fact that not all didactic objectives could be achieved to the required extent. For instance, the design of the interaction options was partly a hindrance to the self-determined adjustment of the difficulty. Nevertheless, the didactic approach and the motivational design could be identified as suitable. For example, the problem presented in the form of an everyday example led to the expected curiosity and motivation by pointing out gaps in knowledge (Table 2 Q15).

#### 3.2.3. Revision and Finalization of the Tool

The revision of the tool was carried out on the basis of the qualitative feasibility study in an iterative process of piloting, analysis and revision. Altogether three revisions were performed: a minor revision after eight interviews with professionals, a major revision after four interviews with laypeople and at the end a second minor revision after interviews with six professionals and four laypeople. The revised and final concept of the tool can be seen in Appendix A. At the moment, the final version of the tool has not been published because we planned to evaluate it in a subsequent research projects, but it can, however, be accessed for scientific interest in German and English language under https://www.leitlinie-gesundheitsinformation.de/RiskTool/englisch/ (accessed on 1 September 2022).

In a major revision, individual contents and the structure of the tool were adjusted. Whereas in the first version of the tool many decisions had to be made by the participants without knowing exactly what content was involved, the revised structure is both more stringent and transparent (Figure 3).

The contents of the tool are now presented in a defined sequence. Animations were replaced, as they were not consistently perceived as helpful. For this purpose, pictograms were used, which were perceived as clear and beneficial for comprehension (Table 3 Q1). Due to the mentioned barriers for accessing the exercises, a mandatory exercise page for calculating the risk reduction has been added (TYK). Since some participants wanted more exercises to improve comprehension, a choice of optional exercises with increasing degrees of difficulty (E-1, E-2) was supplemented with a description of the exercises. These exercises were perceived as helpful, especially due to the increasing difficulty (Table 3 Q2). In addition, it was observed that tasks in the exercise process could be solved more easily and more often without displaying the formula. The section on baseline risk (BS) was completely revised and simplified so that the participants are now able to understand the effect of the basic risk with the help of individual calculation examples and with the same thematic orientation. These changes led to better comprehensibility ratings and participants were now more often able to solve the tasks and understand the meaning of the baseline risk (Table 3 Q3). The tool was also adjusted to suit mobile devices.

Apart from the major revision, minor revisions were carried out in two phases, which were intended to improve the acceptance, applicability, comprehension and didactic achievement of goals. Texts were shortened and sometimes replaced with pictograms. If possible, special terminology was avoided or explained with tool-tips. Option contents were reduced, and important contents highlighted. The tool design and the interaction options were amended, and the reference group highlighted by redesigning the pictograms. The menu navigation and section choices were also revised. These changes resolved the barriers that had occurred and improved clarity. Translation into English was subsequently carried out.

#### 3.2.4. Knowledge Test

Table 4 contains the proportion of correct answers before and after using the tool. Overall, there was no increase in the proportion of correct answers in the before-and-after comparison, except for laypeople in the calculation of the relative risk reduction. In the individual comparison, two people (one professional, one layperson) had better results in the calculation of relative risk reduction and five people (five professionals) had worse results. In the calculation of the absolute risk reduction there were no improvements and eight people (six professionals, two laypeople) had worse results.

## 4. Discussion

A theory-based development process and iterative piloting and revision enabled the development of a tool that allows both laypeople and health professionals to interactively learn how to interpret and distinguish absolute and relative risk reduction in decision-making. As far as we know, this is at the moment the only application of its kind. All in all, the web-based tool is acceptable, applicable and comprehensible and is, in terms of content level, appropriate for both target groups. A stringent structure with reduction in optional content and focus on essential components is beneficial for the conveying of information. Equally beneficial is the way the text difficulty is simplified and the amount of text reduced in conjunction with the use of illustrations, as well as the use of examples related to everyday life and diverse practical applications.

The difference between relative and absolute risk reduction was unknown to a large part of the participants before the study. Only some physicians among the health professionals already knew the distinction in rudimentary form. The topic seems to have been relevant for the majority of the participants and points of contact were found in their own situation. An awareness for the interpretation of risk reduction could be observed, presented therapy effects were questioned more critically and the influence of the presentation of larger, relative effects was recognized.

Regarding the derivation of differences between relative and absolute risk reduction, some participants reported more and some less knowledge gain. In particular, there were difficulties with the calculation in connection with low mathematical skills. The difficulties with the calculation were also apparent in the evaluation of the CHC test items. In total, only two participants improved after completing the tool, while a larger proportion worsened in the calculations. These results are somewhat unexpected, since especially in the revised version of the tool, practice tasks were solved more frequently. These findings are also contrary to those by Jenny et al. [45], who were able to improve the statistical skills of medical students in a 90-min training session. A possible reason for the poorer test results could be the more difficult task in the post-test comparison, since here the calculation had to be made exclusively on the basis of the tabular representations, without assistance in the text of the task, and the outcome to be calculated was less intuitive. In future studies, this should be taken into account when developing the evaluation strategy.

Even if the calculation skills of the participants could not be improved, the awareness for misleading presentation methods and the improvement of the ability to interpret absolute and relative risk reductions is an important step towards enabling participation in health decisions. Including the imparted knowledge in one’s own medical decisions would seem questionable for laypeople, as it suggests that other factors (e.g., gut feeling and attitudes) are more relevant. Moreover, some of them expressed that this tool is only relevant for those who are interested or in special situations. These factors should therefore be taken into account during implementation. Possibilities for integration could be information brochures or information websites of health insurance companies on prevention or early detection services. Here, the tool could be offered as general information or adapted to a specific topic. It could also be used during consultations with doctors or decision coaches [46]. Other areas of application could be the provision of information to patient representatives in guideline processes or studies and representatives of self-help groups. The tool would also be a suitable offer for journalists who report on health topics in order to reduce misleading headlines. An appropriate interpretation of therapy effects is a prerequisite for health professionals to be able to present treatment options in a way that is understandable to lay people and thus enable participation. The tool can support this and be integrated as an independent short training course or as an additional element in existing curricula. Furthermore, it can be used to raise awareness among health professionals who are involved in clinical trials or guideline processes.

The ability to calculate relative and absolute risk reductions is rather secondary in the areas of application mentioned, although it is a requirement in the sense of evidence-based medicine. So far, there are indications that the developed tool cannot improve the calculation skill, even if there is a need for further research due to the mentioned limitations in the interpretation of the results. These explorative findings may provide information for evaluation in a randomized-controlled study. There is also a need for further research on long-term results in terms of the ability to interpret absolute and relative risk reduction and increased awareness.

One of the strengths of the study is its systematic development. This was performed on the basis of a systematic literature search and applying theories about adult education, instruction design, motivation design and multimedia design, giving special consideration to sustaining motivation and preserving cognitive capacity. In the piloting phase, one of the strengths was the combination of the think-aloud method and semi-structured interviews, making it possible to validate statements made during the think-aloud and to deal in more detail with individual aspects. There were, however, limitations. Recruiting laypeople with a low education level was not successful, so that no statement regarding the feasibility could be made for this group. Another point was the fact that the transcriptions were not validated by the participants and even though two participants, who used smartphones or tablets, had no problems, the findings with regard to mobile devices are limited due to the lack of systematic testing.

## 5. Conclusions

A web-based tool for imparting the difference and interpretation of relative and absolute risk reduction to laypeople and professionals in the healthcare system is feasible and can increase the awareness of influences through relative risk reductions in the subjective perception of those concerned. The tool can be used in a variety of ways, ranging from stand-alone use for on-the-job training for professionals, for example, or integration of the tool into websites that inform laypeople about health issues, up to integrating it into existing training courses and curricula, like that of the Network for Evidence-based Medicine [13]. The next step should be a randomized-controlled trial to address the identified research needs. The practical application of the knowledge is relevant for informed decision making, which should also be observed in the further evaluation. Based on the findings, a strategic implementation of the educational intervention is relevant in order to address interested laypeople and health professionals outside educational settings.

## Figures and Tables

**Figure 1 ijerph-19-16086-f001:**
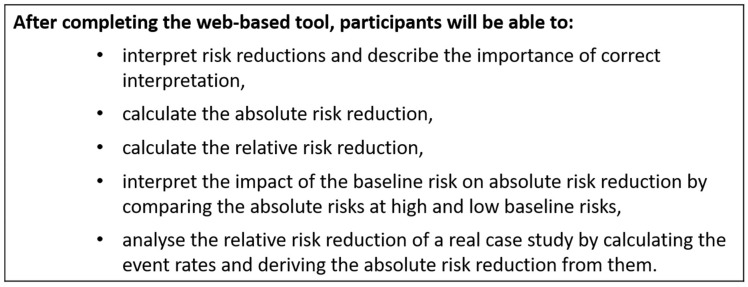
Learning objectives according to Bloom’s Taxonomy.

**Figure 2 ijerph-19-16086-f002:**
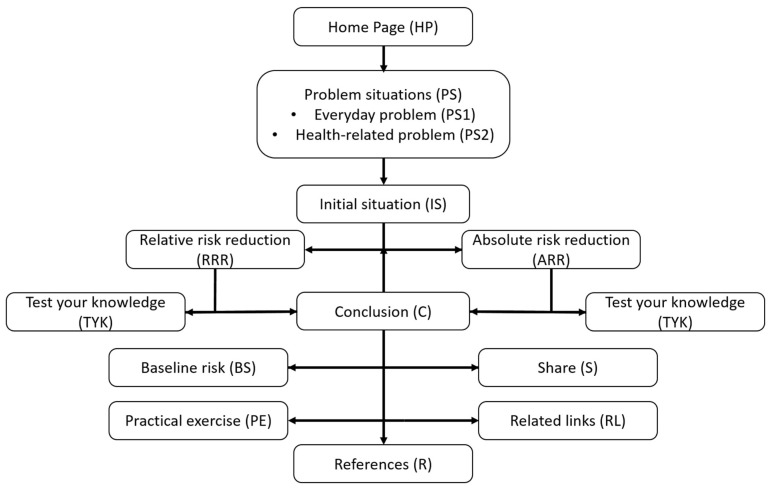
Sequence of topics of the first version of the tool.

**Figure 3 ijerph-19-16086-f003:**
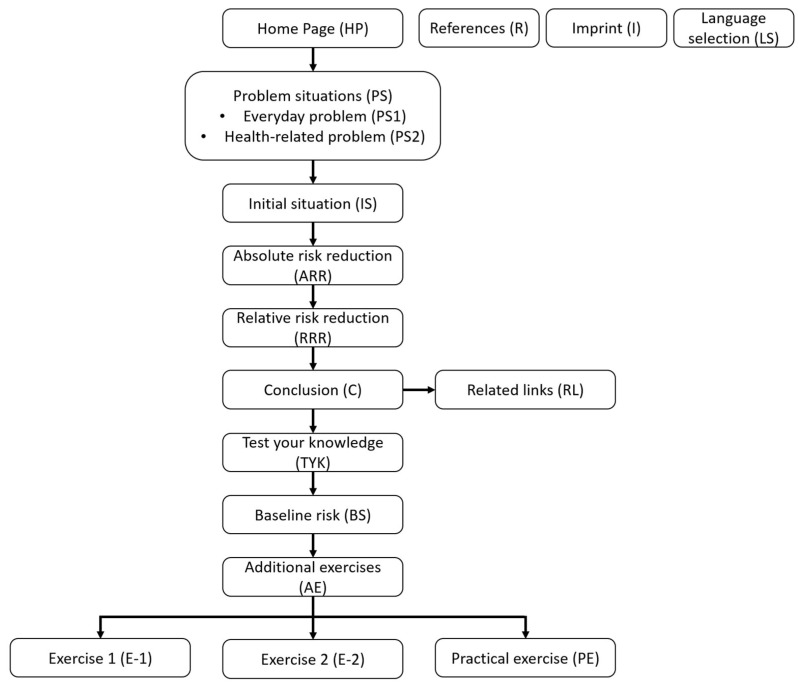
Sequence of topics of the final tool version.

**Table 1 ijerph-19-16086-t001:** Baseline characteristics of participants.

	Laypeople (*n* = 8)	Professionals (*n* = 14)	Total (*n* = 22)
**Age**, mean (R)	34 (21–57)	35 (22–57)	35 (21–57)
**Sex:**			
Female	6	8	14
Non-binary	0	0	0
**Graduation:**			
Secondary school	7	1	8
University degree	1	13	14
**Previous knowledge in statistical basics**	0	10	10
**Mathematical knowledge** *, median (R)	6 (3–7)	5 (1–8)	6 (1–8)
**Computer skills** *, median (R)	8 (6–10)	7 (3–10)	7 (3–10)

R = range; * Scale from 0 to 10; worst to best skills/knowledge.

**Table 2 ijerph-19-16086-t002:** Quotes for characteristic examples regarding feasibility.

**1. Acceptance**	
Satisfaction	
Q1	P3, teacher: *“Yes, no, it’s fun that you are a bit more grounded in your assessment.”*
Emotional reaction	
Q2	L6: *“Oh God, now more calculation exercises are coming. I’m a bit scared of those.”*
Q3	L1: *“Ha, correct! Ha, I’ve got something right, that’s nice. […] That’s what I’ve found out, now I have a sense of achievement.”*
Relevance	
Q4	P1, teacher: *“For the moment, I have identified the following for myself: ok, it’s important that you register that these are just different representations of numbers.”*
Q5	L4: *“[…] I have already thought about getting such an examination [cancer screening] done, and then you would in the end approach the whole thing differently, if you know a bit more about the figures, that you are then more likely to think about it or think differently about it. […]”*
**2. Applicability**	
Structure	
Q6	P5, physician: *“I am an advocate for ‘Practice makes perfect’‚ perhaps you could add one or two more examples or increase the difficulty slightly stepwise. This always helps to get confident about something.”*
Handling	
Q7	P4, teacher: *“Well, then I can see immediately whether I’ve found the right value when it goes green. Really very instructive.”*
Design	
Q8	P2, teacher: *“[…] For me, the question mark was so that I didn’t really take it on board and […] with it I could have calculated faster.”*
**3. Comprehensibility**	
Understanding the content	
Q9	L2: *“[…] You have to see whether it is absolute or relative. And […], that it is two different pairs of shoes in terms of weighting.”*
Q10	P3, teacher: *“[…] Aha, only one person makes a difference. That is not much when one first sees it but, in the end, it is nevertheless really good when one sees the problematic behind it.”*
Difficulty of the exercises	
Q11	P4, teacher: *“Then it was naturally in the example much easier to recognize with the graphic. So it would have been much faster. Well, to be honest, I would have to think again here quite a lot.”*
Text difficulties	
Q12	P5, physician: *“I think the section is good, but I must also say that is something I have to read twice in order to understand it properly, simply because they are complicated matters.”*
Transferability	
Q13	P8, physician: *“[Even if many companies mention only the relative risk reduction at first] I can just assess it. I am more likely to check whether it is relative or absolute.”*
Q14	L6: *“Well, I would be more likely to say that I’d rely on my gut feeling and would not work with a tool."*
**4. Achievement of didactic objectives**	
Q15	P4, teacher: *“[After PS2] I would now, in any case, read the exact explanation about the difference between absolute and relative risk reduction. That makes me a bit curious how it is explained.”*

Q = Allocation of the quotation to the text passage.

**Table 3 ijerph-19-16086-t003:** Quotes for improvements following the revision.

Q1	L6: *“I think it is well explained with the pictures, you can see immediately: this is without early detection, this is with early detection and these are participating persons. […] It’s also motivating to continue because of the way it’s presented.”*
Q2	P11, manager for case management: *“I notice that the more often you do it, the easier it is. […] I can calculate it faster; the practice makes it easier for me […].”*
Q3	P14, nursing student: *"I was then able to calculate everything and the formula was well explained. […] The relative risk reduction is the same for both and in absolute the benefit is smaller for those with a low baseline risk."*

Q = Allocation of the quotation to the text passage.

**Table 4 ijerph-19-16086-t004:** Evaluation of the knowledge test.

Item CHC-Test; Correct Answers (%)	Laypeople (*n* = 8)	Professionals (*n* = 14)	Total (*n* = 22)
t0 RRR (CHC-Scenario 4 Item 16)	2 (25%)	10 (71%)	12 (55%)
t1 RRR (CHC-Scenario 3 Item 5)	3 (38%)	6 (43%)	9 (41%)
t0 ARR (CHC-Scenario 4 Item 17)	4 (50%)	10 (71%)	14 (64%)
t1 ARR (CHC-Scenario 3 Item 6)	2 (25%)	4 (29%)	6 (27%)

## Data Availability

Data are available upon reasonable request.

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
