# Peer review of "Development and Piloting of a Web-Based Tool to Teach Relative and Absolute Risk Reductions"

_ijerph, 2022, doi:10.3390/ijerph192316086_

Round 1

Reviewer 1 Report

This manuscript is a detailed description of developing and piloting a web-based tool related to relative and absolute risk. The need for the tool is not well-established and it is not clear how and under what circumstances it could be used and what it might achieve amongst those who use it.  

Abstract

Line 17: Explored knowledge about what? Please clarify.

Introduction

a)       In general, the introduction does not fully convey the significance and justification for this study. What is the gap? Moreover, why do we need to engage in self-directed learning about this topic? For medical professionals, shouldn’t the differences between relative and absolute risk have been covered in medical training?

b)      Please define relative risk reduction vs absolute risk reduction in the introduction. Also, clarify why relative risk can be misleading.

c)       Please provide some examples of when a layperson would need to understand the difference between relative and absolute risk reduction. I struggle to understand why this topic is critical for health literacy among the general population, especially those that might have low education.

d)      Does the tool attempt to improve a certain category of health literacy (functional, interactive or critical?

e)      Line 29: What kind of study findings and what kind of decision-making?

f)        Lines 44-45: This statement is unclear. Please revise.

g)       Line 51: What does shared decision-making have to do with interpretation of treatment effects?

Methods

a)       Line 95: What is ADDIE?

b)      Line 109: What is ARCS?

c)       Line 142: It is not clear what this phrase refers to: “contents were prepared.” How was content prepared? What content?

d)      Lines 165-209: Please break into multiple paragraphs.

e)      Line 238: Why did you explore efficacy related to knowledge? How does knowledge relate to testing the feasibility of the tool? The stated purpose is to teach the difference between relative and absolute risk reduction. Did you also intend to teach people how to calculate the two types of risk?

f)        Line 273: the categories were deductively created based upon what information or literature?

Results

a)       Lines 289-458: It would seem to me that if you incorporated Table S5, or some version of it, into this section, it would go a long way to making these results easier to read and comprehend.  I do not typically advise this in a qualitative study, but I think this would be a more straightforward approach considering that the great majority of the categories were deductively created, you are not describing deep, qualitative distinctions using your data excerpts, and IJERPH is not a qualitative research journal.

b)      Lines 314-315: How does the first sentence of this quote illustrate an emotional reaction?  Besides being an inaccurate statement, it brings up issues unrelated to acceptance of your tool. Please consider deleting it. “Oh, my God, I’d say mathematics are just not anything for women. […] Well, there again I thought: no, you haven’t understood it anyway, so now you can’t repeat it there either.”

c)       Is there another term to describe the fourth category – tool didactic? It doesn’t make much sense in English. “Didactic tool” makes better grammatical sense, but it still isn’t very descriptive of the category.

Discussion

a)       Few of your participants showed an increase of knowledge (based upon the CHC test). What steps will or did you take to improve this outcome?

b)      In what settings and under what circumstances could the web-based tool be used? How would use of the tool influence health decision-making?

c)       Line 521: What control group?

d)      The Flesch-Kincaid reading level for the post-test scenario 3 is 10.8, which is the equivalent of the third year of high school in the US, or age 16. Please discuss the utility and feasibility of testing this tool with individuals of low education and of even using it (it is current form) in populations with low education or lower reading levels. One of the basic tenets of improving health literacy is to use plain language.

Reviewer 2 Report

Summary

Zacher et al. has written a very detailed description of an online tool to support health care providers and laypeople to learn differences in relative and absolute risk. The tool’s development went through numerous iterations after results from think aloud and interviews. The preliminary qualitative interview data and changes in knowledge after post-interview revisions of the tool are presented within this manuscript.

The work is based on a solid theoretical framework and aside from the lengthy paragraphs that made reading this paper a bit more challenging (cutting into multiple paragraphs would be helpful), the text is overall well written. This reviewer appreciates the details included about the tool’s creation and was glad that the appendices included interview and test questions. What would be helpful would be for the introduction to include some examples of why the differences between relative risk and absolute risk should be better understood. The authors seem to expect that readers of this journal will automatically understand the importance of these values despite their own survey of health professionals indicating a lack of understanding.

Considering the results, this reviewer feels the inclusion of so many participant quotes is unnecessary. Rather, a table of select quotes by category might have better told the research’s story. There was too much text spent on qualitative quotes and yet only a quick summary of the resulting changes that were made as a result of the interviews. It might have been helpful to understand which changes improved what categories. There were also no interviews done after the changes were made to see if these changes resulted in improvements within each category of interest. If this is untrue, why not include a table with quotes acknowledging better understanding and less challenges.

Interesting, the knowledge tests completed after the changes were made showed poor and even poorer performance. Only 2 participants had better test results calculating relative risk reduction and 5 professionals has worse results. No one improved in the calculation of absolute risk reduction and 8 people scored worse. It is not clear if being able to calculate these values is necessary to interpret results using them. While calculation vs interpretations were not really discussed, this reviewer believes that interpretation is of greater importance and therefore this breadth of work still has great value.

This reviewer has the most problem with the Discussion and Conclusion sections. The discussion highlights the strengths of the study but when discussing limitations, focuses on recruitment challenges and lack of verification of transcriptions. They fail to mention the fact that their tool failed to support people’s calculation of both relative and absolute risk. To this reviewer, the emphasis of this paper should be that the tool improves people’s awareness of differences between relative and absolute risk particularly as the tool completely failed at helping both professionals and laypeople alike to calculate relative risk and absolute risk. It seems to this reviewer that participants didn’t know that they needed to know the difference between relative and absolute risk and how this lack of knowledge causes misunderstandings of research findings. Understanding why one needs to know the differences and how better understanding the difference can support medical decision making seems to be the greatest reason to encourage use of the tool. I am not sure this reviewer really received that message from reading the discussion.

Next, the authors indicate that the next step in this line of research would be to do a RCT to both verify the interpretation and calculation. Do the authors think if they give this tool to more people, then more people will have better mathematical abilities and possibly score better on their knowledge tests? It seems to this reviewer; they will get more of the same so why would they propose to pursue such research? This tool seems inadequate for helping people to calculate these values. Discussing its conclusion in a RCT seems to detract from the strengths of the work. Further are there differences between health professionals and laypeople’s use of this tool? How important would offering these laypeople who rarely even see these values be? Rather, might this tool be good to offer research participants or to clinical and research professionals to support their explanation of research findings with their participants and patients? This reviewer sees many uses of this tool that are not included here in the discussion. This reviewer encourages the authors to think bigger.

Finally, this reviewer thinks the first sentence in the title is unnecessary and encourages the authors to use only the section part in the final submission.

Round 2

Reviewer 1 Report

Overall, the revisions much improve the manuscript and the authors have addressed my comments adequately. A few very minor suggestions for the discussion and conclusion:

a) Break up the first paragraph of the discussion into separate paragraphs as there are several topics in there. 

b) line 572 - what are the identified research needs? Long-term uptake of increased awareness and interpretation? 

c) the last couple of sentences of the conclusion are a bit duplicative of the new content in the discussion for how the tool could be utilized.
